# Primary Health Care for Aboriginal Australian Women in Remote Communities after a Pregnancy with Hyperglycaemia

**DOI:** 10.3390/ijerph17030720

**Published:** 2020-01-22

**Authors:** Anna Wood, Diana MacKay, Dana Fitzsimmons, Ruth Derkenne, Renae Kirkham, Jacqueline A. Boyle, Christine Connors, Cherie Whitbread, Alison Welsh, Alex Brown, Jonathan E. Shaw, Louise Maple-Brown

**Affiliations:** 1Wellbeing and Preventable Chronic Diseases Division, Menzies School of Health Research, Charles Darwin University, Darwin, NT 0810, Australia; anna.wood@menzies.edu.au (A.W.);; 2Endocrinology Department, Division of Medicine, Royal Darwin Hospital, Darwin, NT 0810, Australia; 3Northern Territory Department of Health, Darwin, NT 8000, Australia; 4Monash Centre for Health Research and Implementation, School of Public Health and Preventive Medicine, Monash University, Melbourne, VIC 3800, Australia; 5South Australia Health and Medical Research Institute, Adelaide, SA 5000, Australia; 6Faculty of Health and Medical Science, University of Adelaide, Adelaide, SA 5005, Australia; 7Aboriginal Health Domain, Baker Heart and Diabetes Institute, Melbourne, VIC 3004, Australia; jonathan.shaw@baker.edu.au

**Keywords:** type 2 diabetes, gestational diabetes, indigenous health, primary health care, remote health care

## Abstract

Background: Hyperglycaemia in pregnancy contributes to adverse outcomes for women and their children. The postpartum period is an opportune time to support women to reduce cardiometabolic and diabetes risk in subsequent pregnancies. Aims: To identify strengths and gaps in current care for Aboriginal women after a pregnancy complicated by hyperglycaemia. Methods: A retrospective review of the 12 month postpartum care provided by primary health centres in remote Australia in 2013–2014 identified 195 women who experienced hyperglycaemia in pregnancy (gestational diabetes (GDM) (*n* = 147), type 2 diabetes (T2D) (*n* = 39), and unclear diabetes status (*n* = 9)). Results: Only 80 women (54%) with GDM had postpartum glycaemic checks. Of these, 32 women were diagnosed with prediabetes (*n* = 24) or diabetes (*n* = 8). Compared to women with GDM, women with T2D were more likely to have their weight measured (75% vs. 52%, *p* <0.01), and smoking status documented as “discussed” (65% vs. 34%, *p* < 0.01). Most women (97%) accessed the health centre at least once in the 12 month postpartum period but, during these visits, only 52% of women had service provision, either structured or opportunistic, related to diabetes. Conclusion: High rates of dysglycaemia among women screened for T2D after GDM in the 12 month postpartum period highlight the need for increased screening and early intervention to prevent the development of T2D and its complications. Whilst a clear strength was high postpartum attendance, many women did not attend health services for diabetes screening or management.

## 1. Introduction

Diabetes is at epidemic levels in Indigenous communities, contributing considerably to the life expectancy gap between Indigenous and non-Indigenous peoples globally [1]. Hyperglycaemia in pregnancy (which includes gestational diabetes (GDM) and pre-existing type 2 diabetes in pregnancy (T2D)) is increasing, in parallel with the rise in obesity and T2D in women of reproductive age [2]. Hyperglycaemia in pregnancy is associated with poor pregnancy outcomes [3,4]. Long-term sequelae in offspring are also noted [5] and include higher risk of diabetes and obesity later in life [4]. For women with GDM, there is a high risk of developing GDM in subsequent pregnancies and, later, T2D and cardiovascular disease [6]. The burden of disease is greater for Indigenous women, with higher rates of GDM and T2D in pregnancy than their non-Indigenous counterparts [7,8,9,10]. In Australia, Aboriginal and Torres Strait Islander women are 1.5 times more likely to have GDM, 10.4 times more likely to have pre-existing T2D [3], and four times more likely to develop T2D after GDM, compared with non-Indigenous women [11].

In light of the risks associated with hyperglycaemia in pregnancy, postpartum care is essential to addressing postpartum glycaemic status and the long-term risk for both women and their future children. Many women do not present for pre-conception care [12,13] and, hence, the postpartum (or ‘inter-pregnancy’) period may be the only time for opportunistic health messaging prior to future pregnancies. It is an ideal time to support women to reduce their risk of developing T2D and recurrent GDM, or to optimally manage their T2D and other cardiovascular risk factors, with benefits for both women and their future children [14]. However, there are many barriers to achieving this objective. Whilst international guidelines recommend all women with GDM undertake a 75 g oral glucose tolerance test (OGTT) between 6 and 12 weeks after delivery to screen for disorders of glucose metabolism [15], uptake has been low among non-Indigenous [16,17] and Indigenous women [18,19].

Barriers to the delivery of and access to postpartum care have been reported at various levels. Health systems factors include inadequate communication between hospitals and health centres about a diagnosis of hyperglycaemia in pregnancy [16,20] and lack of clarity around who is responsible for postpartum care [21]. At an individual level, barriers for women include the need to fast and the unpleasantness of the OGTT, caregiver demands, prioritising the needs of their family ahead of their own and low perception of the long-term risk associated with hyperglycaemia during pregnancy [22,23,24].

There are further challenges for women living remotely, including accessible and appropriately coordinated health care, the high turnover of health professionals (reducing the establishment of therapeutic relationships and awareness of disease burden) and socioeconomic disadvantage [25,26].

In the Northern Territory (NT), Australia, Aboriginal and Torres Strait Islander peoples comprise 30% of the population with 80% living in remote communities [27]. Standard practice for pregnant women living remotely in the NT is to be transferred from their community to a regional centre at 37–38 weeks gestation to birth in a hospital [28]. During the early postpartum period, most mothers and their babies return to their community and their health care is provided by their local health centres. In the remote setting, there has been limited research into postpartum management after a pregnancy complicated by hyperglycaemia. We aim to identify strengths and gaps in current care for Aboriginal women after a pregnancy with hyperglycaemia to inform the design of a health systems intervention to improve postpartum care and reduce long-term complications.

## 2. Materials and Methods

### 2.1. Study Population

In the NT, remote health care is provided by Aboriginal community controlled health services or the NT Department of Health. The study sites for this review were the remote centres across the NT managed by the NT Department of Health; this included both The Top End (north of Elliot including Tiwi Islands and East Arnhem Land) and Central Australia (including and south of Elliot, including the Barkly region). The study was conducted in accordance with the Declaration of Helsinki and was approved by the Human Research Ethics Committee of the NT Department of Health and Menzies School of Health Research (HREC-2015-2461), and the Central Australian Human Research Ethics Committee (HREC-15-345).

### 2.2. Data Collection

We requested de-identified medical records of all women with a diagnosis of hyperglycaemia in pregnancy (as classified by International Classification of Primary Care (ICPC) codes), who had an antenatal care plan with a birth due date between 1 January 2013 and 31 December 2014. The medical records of 201 women, in 48 different communities, with a diagnosis of hyperglycaemia in pregnancy were assessed. Two women each had two pregnancies and live births during the study period; both pregnancies were included. Four women did not have a confirmed diagnosis of hyperglycaemia in pregnancy and two women appeared to have left the community before birth and did not receive postpartum care in the study communities. These women were excluded from analysis, resulting in 197 entries analysed. Of these women, 75% had GDM, 20% had T2D in pregnancy, no women had type 1 diabetes and 5% had unclear diabetes status during pregnancy.

Within this dataset, information collected included baseline characteristics of women (age, ethnicity, type of diabetes, diabetes medication use during pregnancy and locality), and health care delivered in the 12 months following the baby’s date of birth. Relevant data had predominantly been entered, and was therefore extracted, from care plan templates within the electronic health record (for example a chronic condition care plan) developed in line with best practice guidelines. At the time of this study, guidelines recommended all women have a six-week postpartum check-up and for women with GDM an OGTT be performed [29]. Women with T2D were recommended to commence a chronic condition care plan—involving cardiovascular risk assessment and health checks at least yearly, depending on their level of cardiovascular risk. Women with GDM were recommended to have regular Adult Health Checks—involving a comprehensive health assessment, and yearly glycated haemoglobin (HbA1C) and blood glucose levels [29,30].

### 2.3. Outcome Measures

The outcomes of interest were rates of (1) glycaemic checks performed within the 12 month postpartum period (HbA1C or OGTT), (2) routine six-week postpartum check-ups, (3) commencement of care plans (either Chronic Disease Care Plans or Adults Health Checks) within the 12 month postpartum period, (4) documentation of weight and height measured, body mass index (BMI) calculated, breastfeeding “discussed”, smoking status “discussed” and contraception commenced or “discussed” within the 12 month postpartum period (either prior to, during, or after the six-week postpartum check-up) and (5) recorded outcomes of relevant health checks among women who had the above-mentioned measurements or discussions. To determine whether other attendances were used opportunistically for delivery of diabetes screening and management, information on all other attendances at health centres was noted.

### 2.4. Outcome Variable Definitions

In the 12 month postpartum period, women were classified as having T2D if they had either a diagnostic OGTT (fasting plasma glucose ≥7.0 mmol/L or 2-h plasma glucose ≥11.1 mmol/L), or an HbA1C ≥6.5% (48 mmol/mol). Prediabetes was considered present if fasting plasma glucose was ≥6.1 and <7.0 mmol/L or 2-h plasma glucose was ≥7.8 and <11.1 mmol/L or HbA1C ≥5.7% and <6.5% (39–47 mmol/mol) [31]. Postpartum dysglycaemia was defined as either prediabetes or diabetes. Overweight was defined as BMI 25.0–29.9 kg/m^2^ and obesity defined as BMI ≥30 kg/m^2^. Contraception commenced or “discussed” was defined as any evidence of initiation of contraception, including in hospital immediately postpartum, or any consultation where contraception was documented as discussed. In cases when contraception use changed over the 12 month postpartum period, the initial type of contraception used was recorded as ‘contraception use’. A diabetes-related visit was defined as any visit where, either scheduled or opportunistically, (1) an HbA1C or OGTT was performed, (2) a diabetes-related health professional was seen (diabetes educator, endocrinologist, podiatrist, optometrist), or (3) preventative health advice related to diabetes was given and recorded. A visit where only a random blood glucose was performed was also reviewed, but not included in the main analyses. A visit where only the child was seen was not recorded.

### 2.5. Statistical Analysis

Differences in independent variables and outcomes were determined using Pearson chi square tests for categorical variables, Student’s *t*-test for normally distributed continuous variables and Wilcoxon rank sum tests for non-normally distributed continuous variables. *p* values < 0.05 indicated statistically significant associations or differences. STATA software V 15.0 (Stata Corporation, College Station, TX, USA) was used to carry out statistical analyses.

## 3. Results

Compared to women with GDM, women with T2D were older, more likely to live in Central Australia than the Top End of the NT and had higher postpartum HbA1C and capillary glucose levels (Table 1).

### 3.1. Postpartum Glycaemic Checks

Among women with GDM, 54% (*n* = 80) had either an OGTT, HbA1C or both in the 12 month postpartum period. Thirty-one percent (*n* = 46) of women with GDM had a postpartum OGTT, with the median time to OGTT being 10.4 weeks postpartum [IQR 7.3–19.3]. Twelve percent (*n* = 17) had an OGTT within two months postpartum, as is recommended. Twenty-three percent of women (*n* = 34) had an HbA1C at least 4 months postpartum or later, as recommended. An additional 15% (*n* = 22) had an HbA1C earlier than the recommended 4 months postpartum (if an HbA1C is performed earlier than 4 months postpartum, the results will be influenced by the pregnant state) [32] (Figure 1). There were no differences in age, BMI, or rates of smoking and breastfeeding among women who were and were not screened for dysglycaemia. Women who were screened were more likely to have been using insulin during pregnancy (24% vs. 11%, *p* = 0.04), to have breastfeeding “discussed” (78% vs. 55%, *p* < 0.01), smoking “discussed” (46% vs. 28%, *p* = 0.04) contraception commenced or “discussed” (89% vs. 64%, *p* < 0.01) and have had a six-week postnatal check (89% vs. 71%, *p* = 0.01). However, women who were screened were no more likely to be on a health care plan, compared to women who were not screened.

Of the 54% of women (*n* = 80) with GDM who had either a postpartum OGTT or HbA1C, 31% (*n* = 24) were diagnosed with prediabetes and 10% (*n* = 8) were diagnosed with T2D. Differences between women who did and did not develop dysglycaemia are described in Table 2. Women diagnosed with postpartum dysglycaemia had a higher BMI (29.0 kg/m^2^ (5.3) vs. 24.5 (5.3), *p* = 0.01) and were more likely to be using insulin in pregnancy (34% vs. 17%, *p* = 0.86) though this did not reach statistical significance, compared to women with postpartum normoglycaemia. There was no difference in age between these two groups.

### 3.2. Completion of Six-Week Postpartum Check-Up and Health Care Provided within 12 Months Postpartum

In the 12 month postpartum period, 58% of women (*n* = 115) had a documented specific six-week postnatal check at the health centre, with the median time to review being 7.4 weeks postpartum [IQR 4.6, 10]. Neither age nor diabetes type was associated with the odds of having a six-week postnatal check between women with T2D and GDM, though women living in the Top End were more likely to have a six-week postnatal check (67% vs. 42%, *p* < 0.01) compared to those living in Central Australia.

Within the 12 months postpartum, 25% of women (*n* = 50) were either previously on or commenced a chronic condition care plan and 26% (*n* = 51) had an Adult Health Check. Women on a Care Plan were more likely to be older (30 years (6.5) vs. 28 (5.9), *p* = 0.02) and live in Central Australia (61% vs. 46% in the Top End, *p* = 0.04). Eighty-three percent of women with T2D (*n* = 33) were appropriately either already on a chronic condition care plan, or newly started a chronic condition care plan in the 12 month postpartum period (the remaining 17% were on no care plan) (Figure 2).

Of the 41% of women (*n* = 80) who had a BMI measured, 43% (*n* = 34) were classified as overweight and 29% (*n* = 23) classified as obese. There were no differences in age between women who did and did not have breastfeeding and contraception documented as “discussed”, whilst older women were more likely to have their weight or BMI measured (30 years (6.2) vs. 27 (6.0), *p* < 0.01) and smoking “discussed” (30 years (6.4) vs. 28 (5.9), *p* < 0.01). Women living in the Top End were more likely to have breastfeeding “discussed” (68% vs. 43%, *p* < 0.01), and there were no differences by region in the odds of having BMI or weight measured or smoking and contraception “discussed”.

When comparing women with GDM versus T2D, there were no differences in BMI, although women with T2D were more likely to have had their weight or BMI measured and smoking status documented as “discussed”, and less likely to have contraception commenced or “discussed” There were no differences in whether breastfeeding was “discussed” (Figure 2). In the 67% of women (*n* = 132) who had contraception commenced or “discussed”, the majority (63%) were prescribed Implanon, with no differences in rates of contraception use between women with T2D or GDM (Table 1).

### 3.3. Visits to Primary Health Centre and Reasons for Visit

Among the 97% of women (*n* = 192) who visited the health centre in the 12 month postpartum period, the median number of visits was 10 [IQR 6, 17]. However, 48% of women (*n* = 93) who visited the centre in the 12 month postpartum period had no diabetes-related visits (either an opportunistic or scheduled diabetes-related consultation). Women with T2D were more likely to have a diabetes-related visit than women with GDM, 80% (*n* = 32) compared with 42% (*n* = 62) (Figure 3).

## 4. Discussion

In this study, we reviewed the provision of recommended postpartum care, including routine six-week postnatal check-ups, screening for diabetes and care plans provided to women with hyperglycaemia in pregnancy in remote communities across the NT. This population is at high risk for adverse cardiometabolic outcomes, with a high proportion of women with hyperglycaemia having T2D (20%) compared with GDM, higher than reported nationally (10% in 2014–2015) [33]. In this study, general health centre attendance was high in the 12 month postpartum period. Women with GDM had low rates of postpartum screening for T2D and in those women who were screened, rates of dysglycaemia were high. Women with T2D were more likely to be seen in the postpartum period for reasons related to diabetes, and more likely to have their weight measured and smoking “discussed” compared with women with GDM.

We report low rates of postpartum OGTT screening in women with GDM (31%) at 12 months, although these rates are higher than the 14% reported in Aboriginal women in Far North Queensland [18], and consistent with globally low rates [34]. Some predictors of higher likelihood of postpartum screening have been reported such as older age, lower BMI, Europid ethnicity, insulin use during pregnancy and higher socioeconomic status [35,36,37]. In our cohort of women of mostly non-Europid ethnicity with lower socioeconomic status, we report no differences in age or BMI between women who were and were not screened, although women who were screened were more likely to be on insulin during pregnancy.

The issue of low postpartum OGTT testing raises similar concerns to those raised by low rates of OGTT screening for GDM during pregnancy reported in regional and remote Western Australia [38].

Among women who were screened for T2D, we report a 10% rate of T2D at 12 months postpartum. Chamberlain reports a 22% rate of conversion at 3 years among Aboriginal women in Far North Queensland [11]. In this context, it is critical that we improve adherence to OGTT screening guidelines both during pregnancy and after pregnancy and enable this high-risk group to access timely intervention. It also raises the question as to whether alternate screening strategies that are less cumbersome such as an HbA1C or a risk prediction tool can be further explored [38]. Indeed, whilst reportedly less sensitive [39,40], but cheaper and more convenient, guidelines in regional and remote NT now (after this study period) recommend an HbA1C four months postpartum if an OGTT is not able to be performed [41].

T2D is a serious metabolic condition associated with complications in subsequent pregnancies for the mother and her baby [42]. Contraception discussion and pregnancy planning is critical, particularly given that the degree of hyperglycaemia in early pregnancy correlates with the risk of congenital anomalies in offspring [43]. Among women where contraception was “discussed”, most were using some form of contraception, consistent with a desire for control over family planning and access to postpartum contraception, as reported by Aboriginal women in Queensland [44]. T2D is also associated with other cardiometabolic risk factors and causes numerous complications including cardiovascular disease, chronic kidney disease, neuropathy, retinopathy and peripheral vascular disease [45]. Given this, it is encouraging that women with T2D had relatively high rates of documented weight or BMI measured, or smoking “discussed”. Most women with T2D were also on a chronic condition care plan involving regular cardiovascular risk assessment and health checks. This is reassuring as there is evidence that diabetes care plans result in improved quality of care and cost savings [46]. Specifically, in remote NT communities, it has been shown that a delay from diagnosis to commencement of a diabetes care plan is associated with worse outcomes [47].

Fifty-eight percent of women had a six-week postnatal check, which is substantially better than the 33% documented in 2004–2006 in two remote communities in the NT [48]. Most women in our study (97%) accessed the health centre at some point in the 12 month postpartum period, although there was a significant minority (48%) who accessed the centre multiple times with no documented service provision related to their hyperglycaemia in pregnancy. Most of these women had GDM rather than T2D, perhaps highlighting the lack of awareness of risk associated with GDM. A study of Aboriginal women in South Australia accessing health care in the postpartum period reported women who had GDM were no more likely than women without diabetes to have contact with primary care practitioners in the postpartum period [49]. This suggests that these women were not identified, by themselves or by health professionals, as being at elevated risk.

Health professionals report uncertainty around whose role it is to follow up women with GDM [21] and poor coordination between hospitals and remote health services in the transfer of mother and infant care [50]. Qualitative work with Aboriginal and Torres Strait Islander women and their health professionals in Queensland describes the problem of both women and health professionals not knowing who had GDM and hence who required specific postpartum diabetes care [23]. Studies in women with pre-existing T2D have shown that, despite better glycaemic management during pregnancy, HbA1C values after pregnancy usually worsen [51,52], highlighting the gap in continuous inter-pregnancy care. Continuity of care is also likely impacted by staffing patterns in remote health centres. There is high staff turnover and a reliance on locum or agency staff, reflecting the difficulties in recruitment and retention of health workers in remote communities [53,54].

The Diabetes across the Lifecourse Northern Australia Partnership aims to improve health outcomes for women with hyperglycaemia in pregnancy and their children [12]. This review, coupled with formative work [21,55], will inform the development of a complex health systems intervention in the NT and Far North Queensland which includes improving information management and communication, access to culturally appropriate care and increasing workforce capacity and health literacy of health professionals. In 2017, specific Diabetes in Pregnancy Care Plans were introduced with recalls scheduled in the electronic medical records for postpartum OGTT testing. The timeframe chosen for this baseline study is prior to the introduction of this intervention with the aim to reassess these results after the intervention.

There were several limitations to our study. Data were entered by individual health professionals and thus have potential for human error and missing data. Given the de-identified nature of this data we were unable to perform any cross referencing. We also did not have information on neonatal outcomes. Not all care provided, such as point-of-care HbA1C testing or discussions around smoking cessation, may have been documented; however, even if this is the case, it is important to highlight the need for documentation to understand what sort of health messaging and interventions are taking place. Some discussions may have been documented on the progress notes which would not be captured by an audit of electronic medical records. In addition, women may have moved locations during the period of data collection and may be erroneously recorded as having not received recommended follow-up. The strengths of this study are that we report on real-world rates of postpartum checks and report delivery of key health messages among women who are at high risk for adverse cardiometabolic outcomes.

## 5. Conclusions

In this high-risk population of Aboriginal women with hyperglycaemia in pregnancy, we have described strengths and gaps in postpartum care. Strengths of care reported in this study include high rates of postpartum health centre attendance. Among women with GDM, consistent with other studies, we report low rates of postpartum diabetes screening. Further, in women who had screening, there were high rates of dysglycaemia. These findings highlight a gap and opportunity to work towards increased screening and early postpartum health messaging to prevent diabetes and its complications for these women and future pregnancies.

## Figures and Tables

**Figure 1 ijerph-17-00720-f001:**
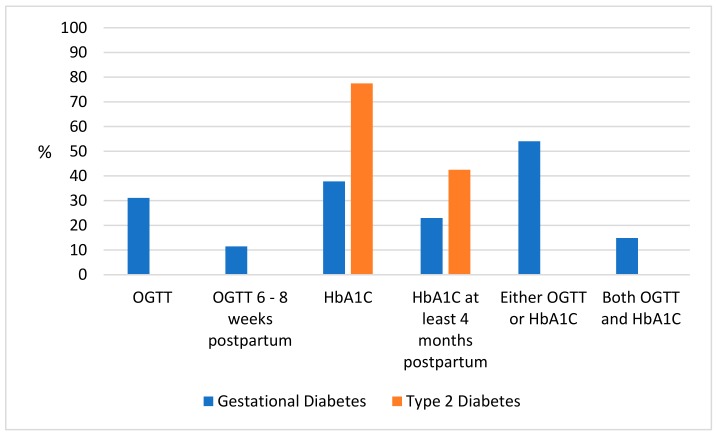
Glycaemic checks performed following hyperglycaemia in pregnancy at any time point in the 12 month postpartum period, in the Northern Territory, 2013–2014, by diabetes in pregnancy type. Data are percentages.

**Figure 2 ijerph-17-00720-f002:**
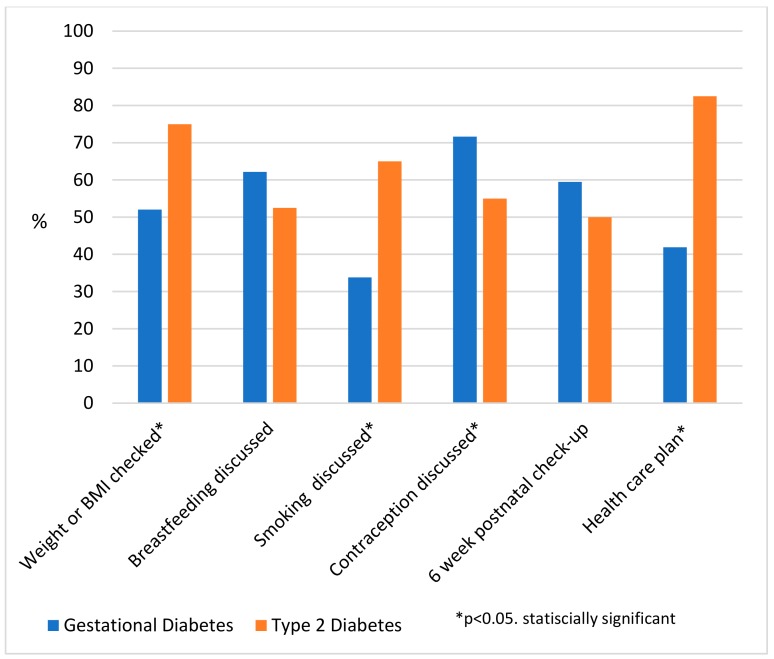
Health care delivered following hyperglycaemia in pregnancy at any time point in the 12 month postpartum period in the Northern Territory, 2013–2014, by diabetes in pregnancy type. Data are percentages. Postnatal Care: Best practice guidelines recommended all women postpartum to have a six-week postnatal check-up. Health care plan: Best practice guidelines recommended women with T2D commence a chronic condition care plan and women with GDM have second yearly adult health checks- both of which form a health care plan.

**Figure 3 ijerph-17-00720-f003:**
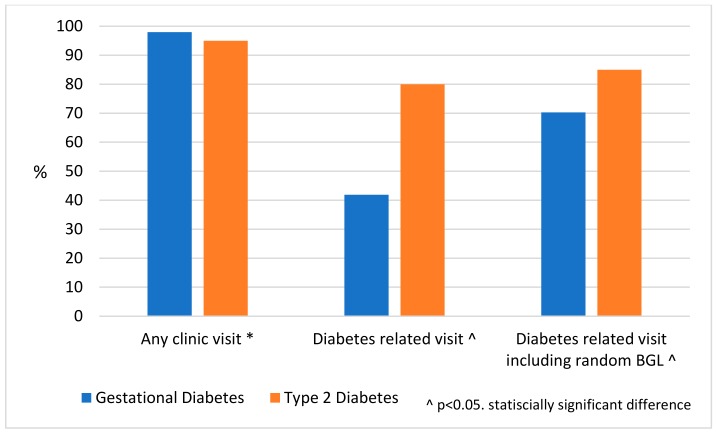
Number of visits to the health centre following diabetes in pregnancy in the 12 month postpartum period in the Northern Territory, 2013–2014, by diabetes in pregnancy type. Data are percentages. * As a percentage of all women in the study. ^ As a percentage of women who had any visit to the clinic in the 12 month postpartum period. BGL, blood glucose level.

**Table 1 ijerph-17-00720-t001:** Characteristics of women following hyperglycaemia in pregnancy and their postpartum results at any time point in the 12 month postpartum period in the Northern Territory, 2013–2014.

Maternal Characteristics	All Pregnancies	GDM	T2D	*p*-Value by Diabetes Type
Number ^a^	197 ^b^	148	40	
Age, mean (SD)	28.7 (6.3)	27.8 (6.2)	32 (5)	<0.001
Ethnicity, n (%)				
Aboriginal	188 (95.5)	139 (93.9)	40 (100)	
Torres Strait Islander	1 (<1)	1 (0.7)	0	
Non-Indigenous	8 (4)	8 (5.4)	0	
Geographic location, n (%)				<0.001
Top End	130 (66)	107 (72.3)	17 (42.5)	
Central Australia	67 (34)	41 (27.7)	23 (57.5)	
**Glycaemic testing results**		n = 46		
		4.4 [4.1, 5])		NA
OGTT fasting mmol/L median (IQR)		5.6 [5.2, 8.2]		NA
OGTT 2 h mmol/L median (IQR)	n = 88	n = 54	n = 31	<0.001
HbA1C % mean (SD)	6.4 (1.5)	6.0 (1.1)	7.3 (1.9)	
HbA1C mmol/mol mean (SD)	46.8 (16.5)	42.1 (11.7)	56.0 (20.3)	
Diabetes treatment, n (%)				<0.001
Diet only	110 (56)	94 (63)	8 (20)	
Metformin only	48 (24)	32 (22)	15 (37)	
Insulin only	8 (4)	5 (3)	3 (8)	
Metformin and insulin	31 (16)	17 (12)	14 (35)	
Weight kg mean (SD) ^c^	n = 113	n = 77	n = 30	0.1927
	72.0 (15.5)	70.9 (17)	75.3 (10.9)	
BMI kg/m^2^ mean (SD)	n = 80	n = 49	n = 26	0.1013
	27.5 (4.9)	26.9 (5.3)	28.9 (4)	
Number of women breastfeeding, n (%) ^d^	n = 117	n = 92	n = 21	0.283
	104 (88.9)	80 (87)	20 (95.2)	
Smoking status ^e^	n = 81	n = 50	n = 26	0.257
	42 (51)	28 (56)	11 (42.3)	
Contraception use ^f^	n = 132	n = 106	n = 22	0.809
**No contraception, n (%)**	9 (8)	7 (6.6)	2 (9)	
Implanon, n (%)	84 (63)	70 (66)	13 (59)	
Other ^g^, n (%)	39 (29)	29 (27.4)	7 (31.8)	

Abbreviations: GDM, gestational diabetes; T2D, type 2 diabetes; SD, standard deviation; OGTT, 75 g oral glucose tolerance test; IQR, interquartile range; HbA1C, glycated haemoglobin; BMI, body mass index. ^a^ All results (glycaemic testing results, weight and BMI measurement, number of women breastfeeding or smoking or using contraception) are the results in those women who had measurements recorded or conversations discussed. ^b^ 9 women unclear whether GDM or T2D in pregnancy. ^c^ Median time weight or BMI measured: 151 days postpartum [IQR 82, 245]. ^d^ Median time breastfeeding discussed: 12 days postpartum [IQR 8, 23]. ^e^ Median time smoking discussed: 180 days postpartum [IQR 103, 267]. ^f^ Median time contraception initiated or discussed: 47 days postpartum [IQR 16, 68]. ^g^ Includes depo provera, tubal ligation, oral contraceptive pill and condoms.

**Table 2 ijerph-17-00720-t002:** Differences between women who did and did not develop postpartum dysglycaemia, among women with GDM who were screened postpartum.

Characteristics	GDM Postpartum Normoglycaemia (*n* = 48)	GDM Postpartum Dysglycaemia (*n* = 32)	Comparison (*p*-Value)
Age years, mean (SD)	28 (6.0)	29 (6.5)	0.456
Prediabetes, n (%)	-	8 (25)	
Diabetes, n (%)	-	24 (75)	
Ethnicity, n (%)			
Aboriginal	46 (96%)	31 (97%)	
Torres Strait Islander	0	1 (3%)	
Non-Indigenous	2 (4%)	0	
BMI, kg/m^2^, mean (SD)	24.5 (5.3)	29.0 (5.3)	0.01
Insulin use during pregnancy, *n* (%)	8 (17)	11 (34)	0.086
OGTT fasting mmol/L, median (IQR)	4.4 (4.1–4.9)	4.8 (4.2–6.8)	0.146
OGTT 2 h mmol/L, median (IQR)	5.5 (4.9–6)	9.1 (8.3–11.4)	<0.01
HbA1C %, mean (SD)	5.5 (0.30)	6.5 (1.3)	<0.01
HbA1C mmol/mol, mean (SD)	36.2 (3.3)	47.5 (13.9)	<0.01

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
