# Peer review of "Primary Health Care for Aboriginal Australian Women in Remote Communities after a Pregnancy with Hyperglycaemia"

_ijerph, 2020, doi:10.3390/ijerph17030720_

Round 1

Reviewer 1 Report

I revised the paper entitled "Primary health care for Aboriginal Australian women in remote communities after a pregnancy with hyperglycaemia". 

Major 

Drug therapy is not reported. It can influence maternal and neonatal outcomes; Newborn outcomes were no reported; Statistical analysis is poor. Authors did not analyzed factors associated with postpartum diabetes care; OGTT data were reported as median and IQR; please analyze with non-parametric test; Outcomes should be stratified by study centres;

Minor

Define OGTT before mention; In discussion, Authors should compare their results with other previous studies.

Reviewer 2 Report

In the manuscript authors assessed the post-partum healthcare system efficacy among women of Aboriginal descent whose pregnancies were complicated by hyperglycemia. The article may be of interest to health professionals providing care to indigenous communities, however, there are minor revisions to be made prior to publication.

Why were women with T1D excluded from the analysis? Please discuss this matter. Do this problem occurs among Aboriginal women? How their health is assessed and screened after the delivery?

In the text "contraception discussed" parameter was not significantly different whereas in the Figure 2 it is. Please correct.

Figure 3. Please explain "BGL" abbreviation.

I would like to see additional table presenting data of 32 women with newly diagnosed pre-diabetes and T2D (age, ethnicity, weight, BMI, OGTT and HbA1c results).

Round 2

Reviewer 1 Report

Authors followed all suggestions. I only suggest to add the lacking of newborn outcomes in limitation section. 

Author Response

Thank you for your comment, 

We have now added the the lacking of newborn outcomes in limitation section to the revised manuscript

Anna Wood